# Emergency Department Visits and Summer Temperatures in Bologna, Northern Italy, 2010–2019: A Case-Crossover Study and Geographically Weighted Regression Methods

**DOI:** 10.3390/ijerph192315592

**Published:** 2022-11-24

**Authors:** Francesco Guolo, Elisa Stivanello, Lorenzo Pizzi, Teodoro Georgiadis, Letizia Cremonini, Muriel Assunta Musti, Marianna Nardino, Filippo Ferretti, Paolo Marzaroli, Vincenza Perlangeli, Paolo Pandolfi, Rossella Miglio

**Affiliations:** 1Department of Public Health, Local Health Authority of Bologna, 40121 Bologna, Italy; 2Department of Statistical Sciences, University of Bologna, 40126 Bologna, Italy; 3Governance of Screening Programs Unit, Local Health Authority of Bologna, 40121 Bologna, Italy; 4Institute for the BioEconomy IBE-CNR, 40129 Bologna, Italy

**Keywords:** microclimate classification, high temperature, case-crossover, geographically weighted regression, emergency department visits

## Abstract

The aim of the study is to evaluate the association between summer temperatures and emergency department visits (EDVs) in Bologna (Italy) and assess whether this association varies across areas with different socioeconomic and microclimatic characteristics. We included all EDVs within Bologna residences during the summers of 2010–2019. Each subject is attributed a deprivation and a microclimatic discomfort index according to the residence. A time-stratified case-crossover design was conducted to estimate the risk of EDV associated with temperature and the effect modification of deprivation and microclimatic characteristics. In addition, a spatial analysis of data aggregated at the census block level was conducted by applying a Poisson and a geographically weighted Poisson regression model. For each unit increase in temperature above 26 °C, the risk of EDV increases by 0.4% (95%CI: 0.05–0.8). The temperature–EDV relationship is not modified by the microclimatic discomfort index but rather by the deprivation index. The spatial analysis shows that the EDV rate increases with deprivation homogeneously, while it diminishes with increases in median income and microclimatic discomfort, with differences across areas. In conclusion, in Bologna, the EDV risk associated with high temperatures is not very relevant overall, but it tends to increase in areas with a low socioeconomic level.

## 1. Introduction

An extensive body of literature has documented the impact of high temperatures and/or heatwaves on cardiovascular, respiratory, and all-cause mortality and morbidity [1,2,3,4,5]. However, to date, the evidence on the association of high summer temperatures with general mortality can be considered consolidated, whereas evidence on other outcomes is still in progress. In particular, relatively few studies have investigated the impact of heat on emergency department visits (EDVs) [6,7], and evidence on vulnerable groups related to temperature EDVs and environmental and spatial effect modifiers remains scant.

EDV is an indicator that lies at a lower severity level than emergency hospitalization or mortality. However, according to some authors, EDV reflects a population’s overall health and should be considered a public health sentinel indicator [8]. In addition, each EDV represents a potentially severe health outcome and places demands on the health care system [9].

A better understanding of the temperature–EDV relationship and the factors that modulate this relationship is necessary to inform targeted interventions. Identifying vulnerable groups, subpopulations, and areas that show a higher risk of EDV due to extreme temperatures can provide helpful information to improve adaptation plans and develop the appropriate strategies to protect subpopulations or tailor geographic-specific programs.

Usually, urban areas tend to have higher temperatures with respect to rural ones because of the urban heat island effect. Likewise, different areas within the same town and at the same moment show microclimatic differences [10]. In addition to air temperature and humidity, factors like vegetation, building density, and composition play an important role in determining an area’s microclimatic characteristics [11,12] in areas where the urban heat island effect is more significant.

As reported in a companion paper [13], the United Nations addressed the Sustainable Development Goals [14] to make cities safer and more inclusive, where number 11 refers directly to sustainable cities and communities. One of the most significant challenges of this Goal is the possibility of designing policies so that the fragile population is protected and has the same opportunities as other citizens, to apply a principle of equity. Indeed, equity is not an option but rather an obligation in a civilized world. Within populations, some vulnerable groups need specific attention or an ensemble of monitoring and care to ensure real inclusion.

Therefore, we conducted a study to assess the effect of heat on EDV and explore the differences across subgroups with different environmental and/or climate characteristics in Bologna, a medium-sized Italian city where a long-standing city-level adaptation plan is in place. In particular, we carried out a case-crossover study on individual data to examine the effect of temperature on EDV and assess whether socioeconomic levels modify the temperature-related EDV relationship according to the deprivation index defined by Caranci et al. [15] and according to microclimatic characteristics as defined by the microclimatic discomfort (MD) index developed by Nardino et al. [13].

In addition, we conducted a spatial analysis on data aggregated at the census block level to study the relationship between environmental characteristics and the EDV rate in days with temperatures beyond a predefined threshold and compare it across space.

## 2. Methods

### 2.1. Setting

Bologna, the main town of the Emilia Romagna Region in northern Italy, has about 390,000 inhabitants. Among the Italian cities belonging to the Covenant of Majors [16], Bologna was the first Italian city to adopt a climate change adaptation plan that was in line with the requirements of the new SECAP [17]. It is considered a municipality where it is possible to introduce improvements targeted at the United Nations SDGs and the new recommendations on ecosystem services produced by the World Meteorological Organization (WMO) [18,19,20] and where there is substantial attention paid to societal factors and human health through the use of urban planning tools that today provide for a microclimatic classification of the city [21]. It is often considered an exemplar for best practices to be applied and scaled on other realities. Especially in the fight against climate change and social inequalities, the city can be considered on the front line of the national panorama regarding the solid efforts and the dedicated resources addressed to these issues.

In Bologna, as in all major Italian cities, a program for the prevention of the effects of heat has been operational since 2004. The program includes a heat warning system, guidelines, the identification of susceptible individuals and a rapid surveillance system of mortality, calls to the emergency service, and EDVs. According to weather condition levels, either just health services or also the general population are warned, and then specific activities are provided to the frailest people identified according to living (alone or cohabiting) and health conditions. At the beginning of the summer, frail residents of Bologna receive letters with recommendations on how to cope with high temperatures. They are called to check their status during heat waves and eventually plan nursing home visits. Throughout the summer period, a telephone number is available for advice and for the activation of some services (free drug and food delivery and transport) in case of special needs.

Bologna has a city centre with buildings dating back to the Middle Ages and Renaissance, narrow roads, arcades, and very little vegetation, and suburbs with more recent buildings, larger roads, and more vegetation.

In Bologna, there are four hospitals with an emergency department. Access is free of charge for all health problems except for nonurgent medical problems considered inappropriate for the emergency departments and that should be taken into charge by the general doctor.

### 2.2. Population

We included all residents in the municipality of Bologna who had at least one EDV in the period 2010–2019 during the summer season from 1 June to 30 September. The study does not consider 2020 events, because we considered this period atypical in terms of access to emergency departments due to the COVID-19 pandemic. Participants were identified using the Emergency Department Archive of all Public Health facilities of Bologna. The archive contains demographic characteristics (date of birth, sex, residence) and the date of the EDV.

Information on each participant was anonymized.

### 2.3. Environmental and Additional Variables

The average daily apparent temperature was considered an exposure variable. Apparent temperature was computed, according to the literature, using the following formula [22,23]:$$Apparent\;temperature=-2.653+0.994\times MT+0.0153\times T_{d}{}^{2}$$
where *MT* is the daily average air temperature while *T_d_* corresponds to the dew point.

To take into account the delay in the effect of temperature, we considered the average exposure between a particular day and the previous day (lag 0–1). The data relating to daily average temperatures and daily average relative humidity were retrieved from the Regional Agency for Prevention, Environment and Energy of Emilia Romagna. They refer to the summer period (from 1 June to 30 September) of 2010–2019. For the project’s aim, the data were collected from a monitoring station located 2 m high in the centre of Bologna.

Information on age, sex, and place of residence (address and district of residence) was collected for each participant. In Bologna, there are six districts: Borgo Panigale-Reno (northwestern part of Bologna), Navile (northern part), Porto-Saragozza (centre and southwestern part), San Donato-San Vitale (centre and northeastern part), Santo Stefano (centre and southern part), and Savena (southeastern part).

The microclimatic characteristics are defined according to Nardino et al. [13], who recently developed an MD index to classify the territory of Bologna and identify areas that are potentially more critical in terms of the urban island effect. As explained in Nardino et al. [13], the MD index took account of the following variables: surface temperature obtained from satellite data during a heatwave that occurred during the summer of 2017; the fraction of vegetation; and building density and the ratio between the average height of the buildings and the width of the road. A normalized microclimatic index was attributed to each block (one or more buildings delimited by four streets) of the town and then was categorized into the following classes corresponding to different levels of thermal discomfort: low (0.15–0.44), mid-low (0.45–0.59), mid-high (0.60–0.72), and high (0.72–1.00) discomfort. According to this classification, the historical centre and some industrial areas are characterized by high values of thermal discomfort, whereas most peripheral areas are characterized by lower values [13] (Figure 1).

The MD index was attributed to each EDV according to the place of residence and to each census block as a mean of all the corresponding values. A census block is a part of the territory according to which each Italian municipality is divided. Bologna Municipality comprises 2333 census blocks, 2085 of these with inhabitants.

The deprivation index defines socioeconomic characteristics. The deprivation index, available for each census block, combines different social and material deprivation traits (low level of education, unemployment, non-home ownership, one-parent family, and overcrowding) relative to 2011 [15]. The deprivation index was attributed to each EDV according to the census block of the patient. According to this index, as shown in Figure 2, the territory is relatively heterogeneous without significant clusters apart from the southern part of the town, with more rich and very rich areas. A higher number of rich areas are also scattered in the city centre than in the suburbs.

We attributed the median income of the statistical area (a level that includes more census blocks) to each patient visit.

For each census block, we also attributed the following information retrieved from Municipality archives: mean age, over 75, and percentage of men. The Appendix A) provides a list of the level of each variable.

### 2.4. Statistical Analyses

#### 2.4.1. Case-Crossover Design

To study the effect of temperature on mortality and to assess whether this effect is modified according to individual-(age, sex, district of residence) or community-level (deprivation and microclimatic index) factors, we applied the case-crossover design. Following this approach, subjects in the case group are treated as control individuals of themselves in a period of nonexposure. In this way, the characteristics of the case and control group individuals are the same, and the study design controls the confounding effect of the constant features of the individuals. Therefore, this design encompasses the need to adjust for time-invariant confounders. Control periods were selected using the time-stratified method in which each case was matched to controls on the same day of the week in the same month and year [24,25].

At first, we explored the concentration–response curve of the relationship between apparent temperature and EDVs. A conditional logistic regression analysis was performed, modelling the exposure variable as a cubic penalized spline [26]. Then, we identified two linear terms and a threshold value for the apparent temperature that leads to an increased risk of EDV over this threshold. Different cut-off values, from 15 °C to 30 °C, were considered, and the threshold associated with the maximum log-likelihood value was chosen. The Likelihood Ratio Test, AIC, and BIC were used to confirm the significance of the cut-off point chosen.

We used a conditional logistic regression model to estimate the risk of EDV in the study population associated with the apparent temperature at lag 0–1. The exposure variable was added to the model with two linear terms associated with the previously computed cut-off point. The first linear term shows the change in the risk of EDV for a 1 °C increase. The second term refers to the additional effect of each 1 °C increase above the cut-off point. The final effect of temperature on EDV over the cut-off point, reported as an OR (odds ratio), was calculated by summing the effect of both terms. The model also included a dichotomous variable that takes into account the temporary population decrease in the summer period (2 weeks in August).

The model was replicated for each potential effect modifier. For the evaluation of the effect modifier, we assumed that the logarithms of the OR were asymptotically normally distributed and independent between the levels of the possible modifiers because these subgroups of the population are disjointed. Under this hypothesis, we tested if the difference in the strata effect (in terms of the logarithm of the OR) is statistically significant based on the analyses of the corresponding confidence intervals [27].

#### 2.4.2. Spatial Analysis

As a secondary analysis, we used data aggregated according to census block to estimate the relationship between the EDV rate of the hottest days and other environmental variables. For each block, we attributed the number of visits to the ED that occurred on the day with an average temperature above the threshold and on the day after and its population size. Then, a classical (global) multivariable Poisson regression model was applied to estimate the increase in the risk of ED visits associated with the MD index, mean age, percentage of men, income, and deprivation index. Subsequently, we used a geographically weighted regression Poisson (GWPW) model [28].

Geographically weighted regression (GWR) models represent a family of regression models in which the coefficients are permitted to vary spatially and focus on local variations. Unlike the global model, the GWR model is a non-stationary linear regression model with an additional function of spatial location [29], where the estimated coefficient parameters can vary over space. GWR models yield as many estimates as the number of the unit of the analyses, with estimates weighted according to the distance from the coordinates of the *i*-th point.

The regression model equation is the following [29]:(1)$$ln(E(IR_{i\;}|X)=\beta_{0}\left(u_{i},v_{i}\right)+\sum_{k}^{p}\beta_{k}\left(u_{i},v_{i}\right)x_{ik}\;\;\;$$
where $IR_{i}$ is the value of the outcome variable at the *i*-th census-block, (*u_i_*, *v_i_*) represent the spatial coordinates of the centroid of the *i*-th census block, and $\beta_{0}$ and $\beta_{k}$ represent the local estimated intercept and effect of *j*-th variable on *i*-th location, respectively. The iterative reweighted least-squares method was used to calibrate the GWPR model using the MGWR software [30].

The idea behind Equation (1) is that nearby data of each location usually possess similar attributes. Thus, choosing an appropriate range (which is referred to as “bandwidth”) makes it plausible to obtain an acceptable local regression.

Using a kernel regression method to calibrate the model to predict smoothed geographical variations in the parameters with a distance-based weighting scheme [30] is a key step in developing the GWPR model. A kernel function defines the reduction of the influence of the weights due to the increase in the distance from the point of interest. The optimal bandwidth was chosen using a bi-square adaptive kernel method, and the calibration process of the estimated model was based on the minimization of the Bayesian Information Criteria.

The variability of the local coefficients over space can be used to examine the plausibility of the stationarity assumption held in the global regression. A Monte Carlo test was performed to determine whether the spatial variability in the local estimates is due to sampling variation.

GWR can be applied only for units of analysis that do not have missing values.

The case-crossover analysis was conducted using STATA (StataCorp, College Station, TX, USA), and the spatial analysis was developed with the MGWR 2.2 (Arizona State University, Tempe, AZ, USA) statistical software. In addition, the georeferencing statistical software QGIS was used to attribute the microclimatic index to each patient residence and to map the EDV rate and local parameter estimates of the GWR by census block.

## 3. Results

The mean apparent temperature during the study period was 24.02 °C (sd 4.01), and the mean temperature was 24.19 °C (sd 3.60). Table 1 presents a summary of the environmental variables considered in this study.

During the study period, 152,440 patients had at least one visit to the emergency department for a total of 261.821 visits. More than 50% of the EDV attendees were 45 years or older and lived in the low-intermediate class of microclimatic discomfort (0.45–0.59), while only 4.13% lived in the highest class of discomfort [13]. The descriptive characteristics of the study population are summarized in Table 2.

### 3.1. Case-Crossover Analysis

After applying the spline function, we identified 26 °C as the threshold point where the risk of dying increases more intensely with a temperature increase. Appendix A in the Appendix A shows the log-likelihood values for different temperature cut-off points.

We found an increase in the probability of EDV equal to 0.43% (OR 1.0043 95%CI: 1.0005–1.0081, *p* = 0.027) for each unit increase of temperature above 26 °C. The variation in EDV below 26 °C is equal to 0.1% (OR = 1.001 95%CI: 0.999–1.003, *p* = 0.166). Table 3 presents the effect estimates according to sociodemographic characteristics and MD class and the *p* value of the z test. The risk of EDV associated with temperature is significant in the 0–5 and the 13–24 age classes (*p* = 0.048 and *p* = 0.034) and for subjects belonging to the highest deprivation index class (*p* = 0.001). As for MD classes, the rise in temperature significantly increased EDV only in the middle-low discomfort area (OR: 1.0062, 95% CI 1.0007–1.0117, *p* = 0.027). Nevertheless, we found no effect modification across variables except for living in one residency district and deprivation index.

### 3.2. Spatial Analysis

Overall, there were 96.331 emergency department visits on days with temperatures over 26 °C and the following days. The median rate was 22.7% (min 0, max 171.4%; IQ range: 17.5–28.9). The map in Figure 3 shows that the territory is very heterogeneous without relevant clusters and that there are more blocks with higher rates in the northern and northeastern parts of the town.

As shown in Table 4, according to the global model, the EDV rate in the days (lag 0–1) with a temperature over 26 °C increases with mean age (+0.55% for each additional year) and with the deprivation index (+5.63% for each additional point) and, conversely, it diminishes with an increase in the MD index: for each increase of 0.1 point in the microclimatic index, the rate decreases by 3.46%, keeping the effect of the other covariates constant. The EDV rate also decreases by 3.29% for an increase of 1.000 euros and by 0.56% for each percentage point of males.

Table 4 summarizes the GWPR estimates and the test results for the spatial variability which show significant spatial variability in the MD index and the median income coefficients but not in the other coefficients. In fact, as Figure 4 and Figure 5 show, in some areas, especially the peripheral ones where there are more deprived areas, the coefficients of the MD index and the median income are positive. However, at the same time, in the global model, they are negative (i.e., the effects of these variables are globally protective).

## 4. Discussion

This large population-based study covers a 10-year period in a setting where a structured heat prevention program is operational.

The first finding of our study is that the risk in EDV changes marginally with high temperatures; we observed that for each unit increase of temperature above 26 °C, there are just 0.4% additional EDVs. This increase is not very relevant from a public health perspective despite the statistical significance, likely due to the high power of the large sample size.

An increased number of EDVs was found in many studies focusing on heatwave periods, for example, the 2006 heatwave in California, U.S. [9], and Paris, France [31], and during the 2011 heatwave in Sydney, Australia [32], and this was confirmed by case-crossover and time-series studies [33,34].

Studies that focus on the association between EDVs and temperature are rarer, and results show some inconsistencies. Hotz et al. [35] found that the risk of overall attendance increased by 1% in London per a 1 °C increase in temperature between 2007 to 2012, with the risk appearing the highest in children. Wang et al. [36] also found an increase in EDVs of a comparable magnitude in 18 sites in China from 2014 to 2017. On the contrary, Tai et al. [37] did not find an effect when considering an increase in the absolute value of the temperature in Taiwan between 2002 and 2007 but did find an effect when considering a temperature change when compared with the previous day and also that only some specialties (trauma) were affected. Basu et al. (2012) [38] found both positive and negative or negligible associations between temperature and EDVs for different circulatory and respiratory diseases in California in the period from 2005 to 2008 [9]. Kingsley et al. reported a slight effect, which found a 1.2–1.4% increase for an increase of 10 °C in temperature in Rhode Island between 2005 and 2012 [39], hence, a magnitude of effect similar to what our study identified.

Besides methodological aspects, heterogeneity in associations may be due to differences in population composition, geographical location, outcome, and population resilience. In particular, the results of our study could be explained by different factors.

Given the available consistent evidence related to heat and morbidity, our study could have lacked showing a relevant association because of the ongoing multilevel climate change adaptation process or because, even if ill, people do not access the emergency department during heatwaves. According to some studies, climate change adaptation is at the root of the decreasing effect of heat on health, which has been documented in recent studies [40,41]. Climate change adaptation consists of physiological, behavioural, infrastructural, and technological adaptation and can involve the individual, community, or population level [40], where home air conditioning has a crucial role [10]. In Bologna, the program for the prevention of heat effects, in place since 2004, could have contributed to protecting the most vulnerable, especially the eldest, and reducing the necessity to attend emergency departments This system has likely contributed to increasing population awareness and modified behaviours.

Furthermore, this system may have directly protected the most vulnerable people thanks to tailored activities for this population group: repeated contact through telephone calls, home visits when necessary, and the activation of specific assistance services. Nevertheless, we cannot assume that the slight increase in EDVs corresponds to a negligible health effect. For example, in an analysis of hospital admissions in 12 European cities, the authors reported that the association between high temperature and cardiovascular admissions tended to be negative and did not reach statistical significance [42]. Some have suggested that these results could indicate that during extreme heat events, people are more likely to die before being admitted to a hospital [42]. This explanation is supported by previous reports of a higher number of deaths outside a hospital than inside a hospital during extreme heat events [34,43]. Other health outcomes should, therefore, be assessed to explore this aspect.

The second finding of our study is that the relationship between temperature and EDVs does not vary with the microclimatic discomfort areas. Although disadvantaged microclimatic areas do not play a role in modifying the risk of EDV, factors such as a lack of green areas, building density, and composition might be compensated by other factors such as building thickness or air conditioning availability or by individual strategies that are adopted during the summer or during heatwaves in particular.

On the contrary, the relationship between temperature and risk of EDV is modified by place of residence. For example, people living in San Donato-San Vitale, a peripheral neighbourhood that is economically more disadvantaged than others, have a greater risk of EDV per each 1 °C increase in temperature. In the same way, the relationship is modified by the deprivation index of the residence: among people living in highly deprived areas, the risk of having an EDV significantly increases by 1.2% for each additional degree. The role of socioeconomic factors as effect modifiers was observed in other studies on temperature and health in relation to mortality [44] and EDVs [34,35,45].

Deprived populations are likely to be more susceptible or less adapted to the heat (maybe due to less thermally efficient housing or the unavailability of air conditioning) and, therefore, more exposed to higher temperatures and less likely to recover from illnesses triggered by heat [35]. Homeless individuals were particularly vulnerable to heat waves in a recent study by Schwarz et al. [46].

It is also conceivable that these results reflect a different attitude in visiting the ED for mild heat-related health problems in people within deprived areas compared to people within less-deprived areas who might first consult the family doctor and do not appear in our statistics [43].

Finally, the spatial analyses show that EDVs are directly associated with deprivation increases and indirectly with median income and microclimatic discomfort. Interestingly, we also found that while the effect of deprivation is substantially homogeneous across Bologna, the effect of microclimatic discomfort varies. In fact, in some areas, it becomes directly associated with EDVs, suggesting that the microclimatic discomfort characteristics of some areas are sometimes compensated but sometimes not, especially in peripheral areas of the town and in more deprived areas. The effect of income is also heterogeneous; in fact, a higher income is not always associated with better health outcomes, suggesting that other unexplored factors are likely to counterbalance the role of this variable. First, though, it must be said that this variable was available only as an area-level variable.

These results should be considered in light of some limitations. Firstly, we did not consider cause-specific EDVs, which could hide some specific heat-related EDVs. Some studies found that some conditions are more frequently associated with temperature-related EDVs than others [37]. Therefore, further research is necessary to explore this aspect.

Secondly, we used a single monitoring station, which may underestimate the effect of temperature. Thirdly, some misclassification might have occurred, as we do not know whether subjects were really exposed to their area characteristics at the time of the event or when they acted as the control. Furthermore, the results of this study largely overlook the conditions of the homeless population, which are extremely difficult to monitor. Yet, this social segment is permanently exposed to the impact of climate change.

## 5. Conclusions

In conclusion, in Bologna, the health risks in terms of EDVs associated with high temperatures are not very relevant, irrespective of microclimatic discomfort. However, they tend to increase in parts of the city with a low socioeconomic level.

Some parts of the city are fragile against heatwaves. However, the adaptation process of the last years, including the policies implemented by the Public Administration to protect the health of the most vulnerable categories, have likely acted as a barrier against the impacts of climate change and have likely helped residents cope with poor microclimates, at least in most parts of the town. Thus, policies to tackle climate change should cover not only urban regeneration actions but also social aspects. Prevention programs of both the Bologna Local Heath Authority and other town institutions should focus more on populations living in deprived areas [47]. This consideration must be evaluated case by case without uncritical generalizations of the specific situation to be solved.

## Figures and Tables

**Figure 1 ijerph-19-15592-f001:**
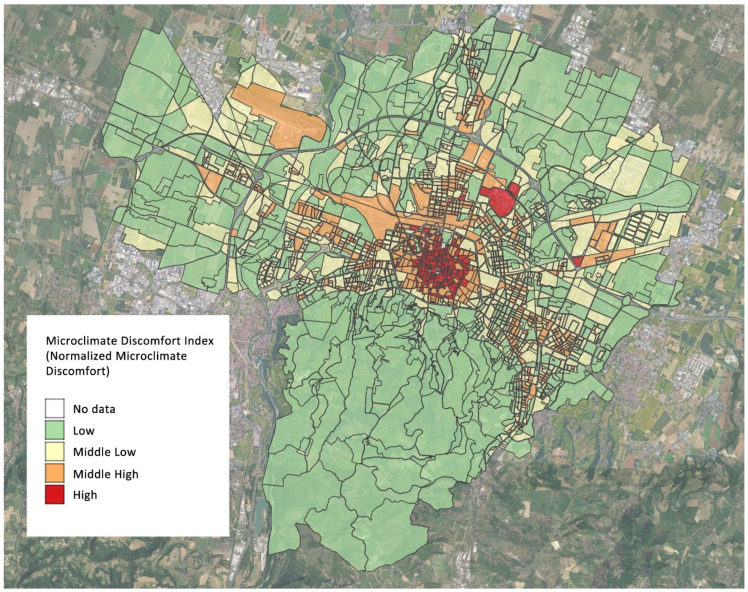
Map of the distribution of the microclimatic discomfort index from Nardino et al. modified [13], Bologna.

**Figure 2 ijerph-19-15592-f002:**
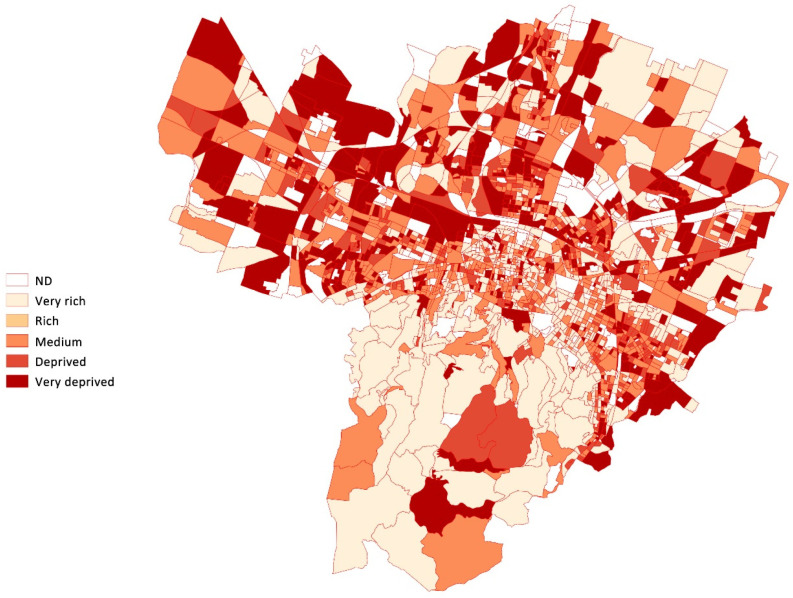
Map of the distribution of the deprivation index, Bologna.

**Figure 3 ijerph-19-15592-f003:**
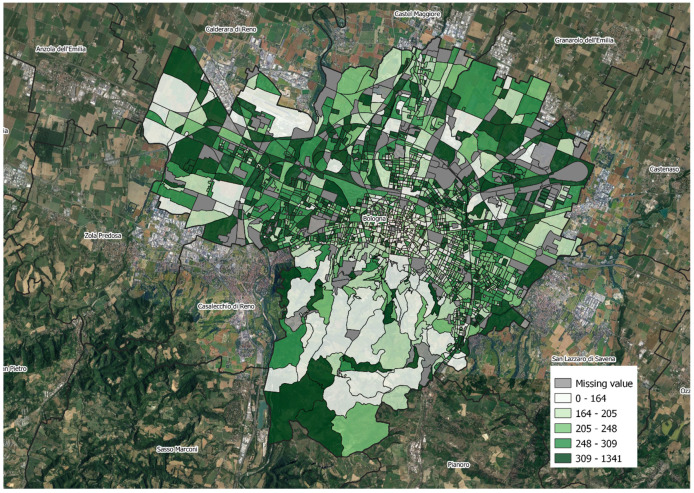
Map of the distribution of the EDV rate ×100, Bologna. The map does not include the block with a rate of 171.4%, because it was derived from a tiny population (*n* = 7).

**Figure 4 ijerph-19-15592-f004:**
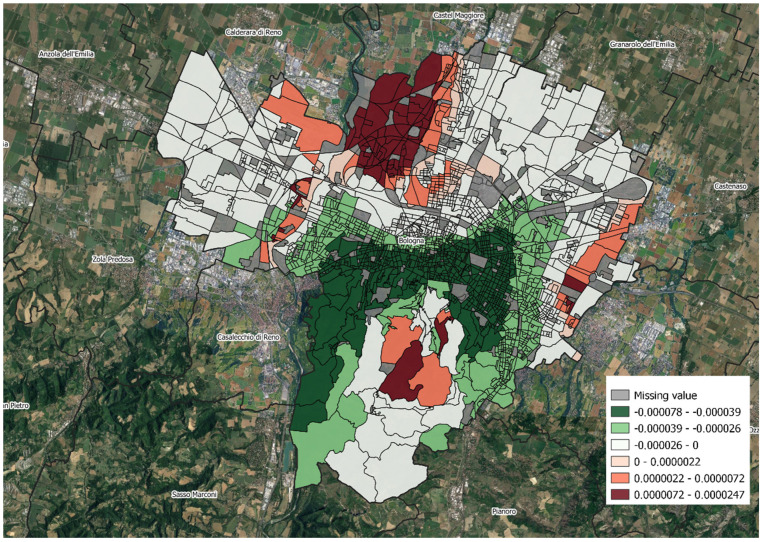
Map of the distribution of the local parameter estimates of GWR: coefficients for median income, Bologna.

**Figure 5 ijerph-19-15592-f005:**
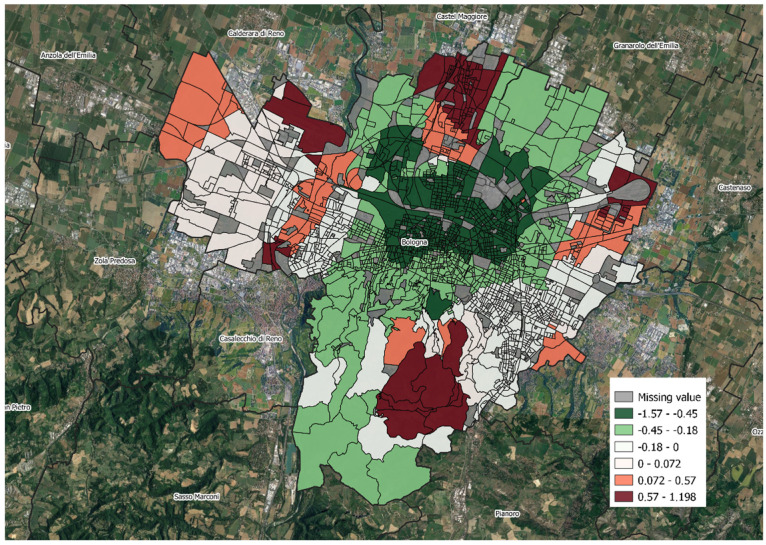
Map of the distribution of the local parameter estimates of GWR: coefficients for microclimatic discomfort index, Bologna.

**Table 1 ijerph-19-15592-t001:** Environmental variable statistics, Bologna 2010–2019.

	Apparent Temperature (°C)	Mean Temperature (°C)	Humidity (%)	Ozone, mcg/m^3^
mean	24.02	24.19	50.51	111.13
sd	4.01	3.60	12.96	31.62
minimum	12.26	13.73	19.00	18.57
maximum	33.19	34.02	95.00	208.63
percentiles				
25th	21.17	21.69	42.00	90.38
50th	24.34	24.56	48.00	112.31
75th	27.08	26.82	58.00	133.00
% missing	0.08	0.08	0.08	5.41

**Table 2 ijerph-19-15592-t002:** Descriptive statistics of the EDV attendees, Bologna 2010–2019.

Variables	*N*	%
**Total**	261,821	100.00
**Age class**		
0–5	21,258	8.12
6–12	12,931	4.94
13–24	19,786	7.56
25–44	65,865	25.15
45–64	53,837	20.56
65–84	60,835	23.24
>84	27,309	10.43
**Gender**		
Male	119,476	45.63
Female	142,345	54.37
**District of residence**		
Borgo Panigale—Reno	46,753	17.86
Navile	47,795	18.25
Porto—Saragozza	42,552	16.25
San Donato—San Vitale	49,446	18.89
Santo Stefano	34,856	13.31
Savena	40,419	15.44
**Deprivation index**		
Low (−4.8–−0.97)	41,705	15.93
Middle-low (−0.96–−0.30)	47,170	18.02
Middle (−0.29–0.26)	46,283	17.68
Middle-high (0.27–1.00)	49,233	18.80
High (1.01–24.61)	77,430	29.57
**MD class**		
Low discomfort	55,696	21.27
Middle-low discomfort	125,223	47.83
Middle-high discomfort	69,815	26.67
High discomfort	11,087	4.23

**Table 3 ijerph-19-15592-t003:** OR (odds ratio) of EDV per 1 °C increase above 26 °C in mean apparent temperature (lag 0–1) and 95% confidence interval (CI) in the study populations according to demographic characteristics and microclimatic class and *p* value of the z test.

	OR	95%	CI	*p*	z Test, *p*
**Total**	1.0043	1.0005	1.0081	0.027	
**Age class**					
0–5	1.0139	1.0001	1.0279	0.048	ref
6–12	0.9943	0.9765	1.0124	0.537	0.091
13–24	1.0154	1.0012	1.0298	0.034	0.884
25–44	1.0008	0.9933	1.0084	0.831	0.104
45–64	1.0065	0.9982	1.0149	0.125	0.370
65–84	0.9993	0.9915	1.0071	0.852	0.071
>84	1.0084	0.9968	1.0200	0.156	0.548
**Gender**					
Female	1.0042	0.9990	1.0094	0.110	ref
Male	1.0044	0.9988	1.0101	0.125	0.951
**District of residence**					
Borgo Panigale—Reno	1.0023	0.9934	1.0112	0.619	0.990
Navile	1.0068	0.9979	1.0157	0.136	0.769
Porto—Saragozza	1.0080	0.9985	1.0175	0.098	0.922
San Donato—San Vitale	1.0086	0.9999	1.0175	0.054	ref
Santo Stefano	0.9926	0.9821	1.0031	0.167	0.022
Savena	1.0048	0.9951	1.0146	0.336	0.564
**Deprivation index**					
Low (−4.8–−0.97)	1.0062	0.9967	1.0159	0.203	0.342
Middle-low (−0.96–−0.30)	0.9967	0.9878	1.0057	0.476	0.009
Middle (−0.29–0.26)	1.0033	0.9943	1.0124	0.470	0.138
Middle-high (0.27–1.00)	0.9985	0.9897	1.0073	0.736	0.019
High (1.01–24.61)	1.0120	1.0050	1.0190	0.001	ref
**MD class**					
Low discomfort	1.0023	0.9940	1.0106	0.586	0.581
Middle-low discomfort	1.0062	1.0007	1.0117	0.027	0.852
Middle-high discomfort	1.0019	0.9946	1.0093	0.615	0.547
High discomfort	1.0081	0.9896	1.0269	0.395	ref

**Table 4 ijerph-19-15592-t004:** Relationship between EDV rate during days with temperature >26 °C and census block characteristics. Estimates of Poisson and GWRP model. * Some census blocks were not considered because of missing data.

	Poisson Regression Model—Global Model (n Census Block = 2014) *				Summary of GWPR Estimates (n Census Block = 2014) *						Montecarlo Test for Spatial Variability
						Coefficients		% Change in EDV			
Variables °	Coefficients	SE	*p*	% Change in EDV (95%CI)	Min	Median	Max	Min	Median	Max	
Intercept	−0.539	0.0767	<0.001		−2.190	−0.803	1.171				0.709
Mean age	0.0055	0.0008	<0.001	0.55 (0.39–0.70)	−0.020	0.008	0.019	−2.01	0.80	1.96	0.327
Male percentage	−0.0056	0.0008	<0.001	−0.56 (−0.71–−0.41)	−0.033	−0.005	0.007	−3.28	−0.47	0.69	0.446
Median income (×1000)	−0.0335	0.003	<0.001	−3.29 (−3.64–−2.95)	−0.078	−0.031	0.025	−7.52	−3.01	2.50	0.010
Deprivation index	0.0547	0.0011	<0.001	5.63 (5.39–5.86)	0.018	0.058	0.092	1.79	5.94	9.59	0.724
MD index (×0.1)	−0.0353	0.00290	<0.001	−3.46 (−4.01–−2.92)	−0.157	−0.024	0.120	−14.52	−2.36	12.73	<0.001

* Some census blocks were not considered because of missing data. ° All listed variables were included in the model simultaneously.

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
