# Peer review of "Emergency Department Visits and Summer Temperatures in Bologna, Northern Italy, 2010–2019: A Case-Crossover Study and Geographically Weighted Regression Methods"

_ijerph, 2022, doi:10.3390/ijerph192315592_

Round 1

Reviewer 1 Report

1. At the beginning of the Methods section (l 83-89), it is said that the site of the study (Bologna) is at the forefront in applying best practices in public health. Does this claim need to be substantiated by evidence/references?

2. Given that there has been a long-standing city-level adaptation programme (since 2004) on the effects of heat on health, the authors need to provide more justification for why they chose this population to assess risk and effects, when vulnerability might have already been reduced. Presumably, the evaluation might have been more valuable if this population was compared with a matched city where there was no such adaptation programme already in place?

3. Given the finding of only marginal change in EDV associated with high temperature, it seems doubtful that the authors had much scope to explore the relative impact of microclimatic discomfort in evaluating that association. This explanation for qualifying a lack of association could be emphasised more clearly in the Summary.

4. The finding (l 381- and in Abstract) that EDV is directly associated with deprivation index and indirectly with median income is an important finding (although, as the authors make clear, not new) - it would be helpful to have a simple, succinct, expression of the size of this effect in the Abstract and discussion.

5. The results show that the risk of EDV associated with temperature is statistically significant (only) in the 0-24 age class. Would the authors agree that the lack of an effect in the elderly, usually a particularly vulnerable group, might again be explained by the success of the established heat-health intervention programme? Is it possible to disaggregate the 0-24 age class further e.g. was the greatest effect seen in young children? Can any of the differential effects between ages be used for more specific, discussion of the relative success of the heat-health adaptation plans already in place?

6. This manuscript is submitted as part of a special issue on urban resilience and population health, and several of the authors are from the Local Health Authority. It would be of interest to the readership to know how this Local Health Authority is now acting on the results obtained - both in terms of local action and in sharing the lessons of good practice with other Local Health Authorities.

Author Response

Dear Reviewer, please look at the attached file.

Reviewer 2 Report

I read with interest this paper regarding the association between climate and Emergency department visits.

The paper is well written and presented, however, some issues could be addressed to improve its readability

1) The math behind the study is very complex, still, some points could be clarified. The authors suggest that starting from 26°C the rates of ED access increase, and the average annual temperature is 24°C. The cut-off point selection appears arbitrary and is claimed by the authors as the result of a non-shown analysis. The authors could provide a table (at least in the supplementary materials) with a table resuming this analysis and showing the changes for each threshold.

2) While the correction for the micro-climatic characteristics of the patient's residence appears appropriate and very well carried out, I have some concerns about the relevance of the "census" correction for this analysis.

The authors should clarify why this is important in this context.

3) Since the analysis is corrected for the residence, it should take into account the overall numerosity of the sample in each area (i.e. the inhabitants for each zone of the city). Indeed, the crude number of inhabitants is the major driver of ED access for each zone.

4) Finally. The authors do not take into account the seasonal variation of temperature in the city area. It appears that an average temperature above 26°C could be present just in the summer in that region. However, this variation could be due to many other unexplored factors, like for instance the increase in the overall population due to the increased number of tourists in the area, or the reduction of territorial medical activities that could push the people to the ED.

The authors should divide the analysis for the season to verify if the tendency is present in case of climatic variations regardless of the season.  

Author Response

(The authors gave the same response as above.)

Reviewer 3 Report

The manuscript aimed to evaluate the effect of heat on Emergency Department Visits (EDV) in Bologna during summer 2010-2019 and the effect-modification by individual and environmental factors. There are few papers addressing the effect of heat on morbidity and evaluating the role of both individual- and area-level conditions that might identify specific groups or areas more vulnerable to the effect of heat at intra-city level. The manuscript is well written, concise and informative. The data sources come from public datasets for a variety of co-variates and potential confounders at ecological level. The methods are adequate for main analysis (case-crossover analysis) and the authors conducted additional spatial analysis for assessing the effect modification of environmental variables at area-label, which complement adequately the main analysis. The results are concise and well presented and the discussion section include limitations as well as explanations of potential mechanisms for the association and effect-modification effect found.

In line 341 authors mention “climate change consists of physiological, behavioral….” And maybe the concept is not climate change but climate change adaptation.

Author Response

(The authors gave the same response as above.)

Round 2

Reviewer 2 Report

The authors answered satisfactorly to my comments.